# Green Tea Suppresses Brain Aging

**DOI:** 10.3390/molecules26164897

**Published:** 2021-08-12

**Authors:** Keiko Unno, Yoriyuki Nakamura

**Affiliations:** Tea Science Center, University of Shizuoka, 52-1 Yada, Suruga-ku, Shizuoka 422-8526, Japan; yori.naka222@u-shizuoka-ken.ac.jp

**Keywords:** arginine, brain aging, catechin, epigallocatechin gallate, green tea, stress reduction, theanine

## Abstract

Epidemiological studies have demonstrated that the intake of green tea is effective in reducing the risk of dementia. The most important component of green tea is epigallocatechin gallate (EGCG). Both EGCG and epigallocatechin (EGC) have been suggested to cross the blood–brain barrier to reach the brain parenchyma, but EGCG has been found to be more effective than EGC in promoting neuronal differentiation. It has also been suggested that the products of EGCG decomposition by the intestinal microbiota promote the differentiation of nerve cells and that both EGCG and its degradation products act on nerve cells with a time lag. On the other hand, the free amino acids theanine and arginine contained in green tea have stress-reducing effects. While long-term stress accelerates the aging of the brain, theanine and arginine suppress the aging of the brain due to their anti-stress effect. Since this effect is counteracted by EGCG and caffeine, the ratios between these green tea components are important for the anti-stress action. In this review, we describe how green tea suppresses brain aging, through the activation of nerve cells by both EGCG and its degradation products, and the reductions in stress achieved by theanine and arginine.

## 1. Introduction

Since the aging population is growing worldwide, the prevention of brain aging is a universal problem. In Japan, in particular, the number of elderly people is increasing rapidly, with the highest percentage of elderly people in the world. According to data collected in 2012 by a research group from the Ministry of Health, Labor and Welfare of Japan, over 70% of women aged 95 years and over suffer from dementia. Similarly, over 50% of men aged 95 years and over suffer from dementia. Besides genetic factors, dementia is greatly affected by brain aging, environmental factors, and lifestyle-related factors. Improving these aspects and mitigating age-related changes are believed to help to delay the onset of dementia. With age, various changes, such as arteriosclerosis, climacteric disorder, diabetes, hyperlipidemia, kidney disease, hypertension, obesity, and poor memory, commonly occur. However, these age-related changes have reportedly been improved or counteracted in people who followed advice pertaining to exercise and diet [1]. Ingesting green tea is one such recommendation. In fact, green tea consumption has been epidemiologically shown to be inversely associated with both male and female heart disease mortality and male cerebrovascular and respiratory mortality [2]. In addition, there are accumulating reports that the ingestion of green tea is effective in preventing dementia [3,4,5,6,7]. In order to elucidate the effect of green tea, studies mainly focusing on the antioxidant effect of epigallocatechin gallate (EGCG) have been conducted in vivo and in vitro [8,9,10,11]. The effects of EGCG are often strong, but the effects of green tea cannot always be explained by the effects of EGCG alone. It is also necessary to investigate the interactions among other green tea components, such as caffeine, theanine, and arginine. In addition, the involvement of green tea metabolites must be studied too. Detailed studies have been conducted on the metabolites of green tea catechins [12,13,14], and research on their biological effects has been undertaken in recent years [15,16,17,18,19,20,21]. It should be noted that the metabolites produced by the gut microbiota have important effects on our health.

Theanine, the most abundant free amino acid in green tea, has been found to relieve stress and have a relaxing effect [22,23,24,25,26]. In addition, arginine, which is the second most abundant free amino acid in Japanese green tea, has an anti-stress effect similar to theanine [27,28]. However, since the stress-reducing effects of theanine and arginine are counteracted by EGCG and caffeine, the composition ratio of these components is important for the anti-stress effect of green tea [27,29]. Stress is deeply involved in the development and progression of many diseases, such as asthma, hypertension, peptic ulcers, obesity, and diabetes. The brain is also subjected to significant stress. Even healthy people undergo atrophy in the brain when they are repeatedly subjected to intense stress—for example, as a result of bereavement, job loss, accidents, or major natural disasters [30,31,32]. It has also been reported that brain atrophy occurs in abused children [33,34,35]. Furthermore, many people are more stressed than before due to the current coronavirus pandemic [36,37]. The stress-reducing effect of theanine and green tea is thought to contribute to the maintenance of mental health and the control of brain aging in many people.

In order to clarify the functionality of foods, we must examine the interactions among food components and the actions of metabolic decomposition products. In this review, we focus on the effects of green tea components—catechins, EGCG metabolites, theanine, and arginine—on brain function. In addition, we explain the anti-stress and anti-depressant effects of several types of green tea.

## 2. Catechins

Among the tea catechins, epicatechin was first isolated from green tea extract by Michiyo Tsujimura in 1930 [38]. Catechin can take four different structures: (+) catechin, (−) catechin, (+) epicatechin, and (−) epicatechin; the (−) epi-form is most common in green tea. In addition, gallic acid is contained in catechins and gallocatechins, and (−) epigallocatechin gallate (EGCG) is the most abundant catechin in green tea. (−) Epigallocatechin (EGC), (−) epicatechin gallate (ECG), and (−) epicatechin (EC) are also found (Figure 1). Unlike normal flavonoids, catechins rarely exist in nature as a sugar-bound glycoside. Although catechins are also found in arboreal fruits such as cacao, grapes, apples, and peaches, their main components are free catechins, such as (+) catechin, (−) epicatechin, and (−) epigallocatechin, that do not contain gallic acid. Therefore, gallate-type catechins such as EGCG and ECG are unique to tea leaves.

### 2.1. Lifespan

Brain aging does not always coincide with body aging, but lifespan is an important marker of body aging. It has been confirmed that green tea or EGCG has a life-prolonging effect on nematodes, flies, and mice [39,40,41]. According to the results of an epidemiological survey of approximately 91,000 Japanese people aged between 40 and 69, who were followed for 18.7 years, all-cause mortality was significantly reduced in those who drank five or more cups of green tea every day, compared to those who drank less than one cup/day [2]. Similarly, elderly Chinese men who consumed tea almost every day showed a 10–20% lower risk of death compared to their counterparts who seldom consumed tea [42].

The relationship between the intake of green tea catechins and lifespan was investigated using senescence-accelerated model mice, SAMP10/TaSlc (SAMP10), with a short lifespan. The median survival time (MST) was significantly prolonged when 1 mg/kg (equivalent to one cup of green tea per day in humans) was taken daily [41]. Even when 5 to 60 mg/kg of green tea catechins was taken, MST was prolonged, although not significantly, when compared with mice consuming no green tea catechins [41]. It thus seems important to drink several cups of green tea at least daily, but not to consume large amounts of catechins.

### 2.2. Cognitive Function

Green tea intake has been reported to prevent cognitive decline [3,4,43,44]. In experiments using SAMP10, we found that the daily ingestion of green tea catechins suppressed brain dysfunction in aged mice [45,46]. To clarify the effect of green tea catechins on the suppression of cognitive impairment with aging, the effect of the start time of green tea catechins intake was investigated in aged SAMP10 mice. Six- or nine-month-old SAMP10 mice began drinking water containing green tea catechins until they were 11 months old [47]. In SAMP10 mice, six and nine months of age are considered an adult and middle-aged, respectively. Mice that received green tea catechins from 6 to 11 months had significantly higher memory acquisition capacity, as examined by the passive avoidance trials, than control mice of the same age who did not consume green tea catechins. Consuming green tea catechins from 9 to 11 months also tended to improve learning ability. The daily ingestion of green tea catechins has been suggested to suppress brain dysfunction with aging. Additionally, even if ingestion started from middle age, this appeared to be more beneficial than not consuming it at all.

Next, the dose-dependency of the effect of green tea catechins on cognitive function was examined. A significant effect on memory acquisition was observed at 1 mg/kg or more of green tea catechins, as assessed by the passive avoidance test, and the most effective dose was observed to be 15 mg/kg [41]. Long-term memory retention was significantly higher in mice consuming green tea catechins daily at 60 mg/kg. Spatial working memory measured using a Y-maze was significantly increased in mice that ingested green tea catechins at 30 mg/kg and over [41]. In other words, green tea catechins’ suppression of cognitive decline was dose-dependent, with a minimum required dose of 1 mg/kg in mice (equivalent to one cup of green tea in humans).

Comparing the effects of catechins between EGCG and EGC, EGCG suppressed the decline in cognitive function, but no effect was observed with EGC at the same concentration [48]. The difference between the two is epitomized by the presence or absence of a galloyl moiety, but gallic acid (GA) itself had no effect [48]. As for the action of catechins, the importance of antioxidative activity has been highlighted. In fact, in mice ingesting the green tea catechins EGCG or ECG, lipid peroxidation in the cerebral cortex was significantly reduced compared to the control group, indicating that the antioxidant capacity was not significantly different between EGCG and EGC [48]. In other words, it is difficult to completely explain the difference between EGCG and EGC by antioxidative activity alone, on the suppression of cognitive decline.

### 2.3. Absorption and Metabolism of Catechins

When EGCG is orally ingested, a small amount is taken up from the small intestine and enters the systemic circulation with no degradation [49,50], and only a trace amount remains eight hours after ingestion. On the other hand, most EGCG is decomposed into EGC and GA by the intestinal microbiota [51]. Degalloylated catechins have been reported to be further degraded in the large intestine, converting into 5-(3′,5′-dihydroxyphenyl)-γ-valerolactone (EGC-M5) (Figure 2), 5-(3′,4′-dihydroxyphenyl)-γ-valerolactone, and 5-(3′,4′,5′-trihydroxyphenyl)-γ-valerolactone as the major degradation products more than eight hours after ingestion [13,14,52]. These valerolactones are readily absorbed into the large intestine and subsequently metabolized and distributed [13]. The conjugated forms of these catechins and metabolites, such as sulfates (EGC-Mg-Sul) and glucuronides (EGC-M5-GlucUA), have been found in plasma and urine [13]. GA becomes a glucuronide conjugate (PG-GlucUA) via pyrogallol (PG) [51] (Figure 2).

### 2.4. Blood–Brain Barrier Permeability of Catechins and Their Degradation Products

Since catechins and their degradation products need to cross the blood–brain barrier (BBB) in order to act directly in the brain, the permeability from the vascular side to the brain parenchyma side was measured using an in vitro BBB assay kit (RBT-24, Pharmaco-Cell Company Ltd., Nagasaki, Japan). The permeability of EGCG was 4% and that of EGC was 5% in 30 min [17,48] (Table 1). These values are lower than the permeabilities of caffeine and GA, but catechins such as EGCG and EGC undoubtedly reached the brain parenchyma. Using a similar in vitro BBB model, the transfer rates of (+) catechin and (−) epicatechin have been reported to be 7.4% and 15.4% in 1 h, respectively [53]. This indicates that the difference in the configuration of the hydroxyl group at the 3-position of the flavan structure greatly affects BBB permeability. EGCG has a large galloyl group at the 3-position, but the high permeability of GA is considered to enhance that of EGCG to a level similar to EGC. EGC-M5 showed slightly higher permeability than EGCG and EGC. EGC-M5 conjugates were less permeable than EGC-M5, while EGC-M5-Sul showed slightly higher values than EGCG [17]. These data indicate that the effects of both catechins and their degradation products in the brain should be noted. On the other hand, when EGC and GA were present together, the permeability of EGC was considerably reduced as compared with EGC alone. Further research is needed to evaluate whether such competitive effects actually occur in the brain.

### 2.5. Effect of Green Tea Catechins on SH-SY5Y Cells

SH-SY5Y, a human neuroblastoma, was used to investigate the action of green tea catechins that entered the brain parenchyma. When SH-SY5Y cells were treated with 50 nM of catechins, the neurites, which are markers of nerve cell differentiation, became longer, and their number increased. The action of EGCG was stronger than that of EGC and GA (Figure 3) [48]. The effect of EGC-M5 on SH-SY5Y cells was the second strongest after EGCG [17]. EGC-M5-Sul was also found to have a neurite-stretching effect [17]. Interestingly, it was reported that when 5-(3′,4′-dihydroxyphenyl)-γ-valerolactone was administered to rats, the sulfate-conjugated 5-(3′,4′-dihydroxyphenyl)-γ-valerolactone was detected in the brain, but not in the form of aglycones [15]. Therefore, EGC-M5-Sul may play an important role in the brain as an EGCG degradation product.

### 2.6. Altered Gene Expression Due to Green Tea Catechin Ingestion

In order to elucidate the mechanism by which the intake of green tea catechins suppresses the decline in brain function, we investigated which genes’ expression was changed in the brain. The hippocampus of SAMP10 mice treated with green tea catechins for one month displayed significantly increased expression of some of the immediate early genes (IEGs), such as nuclear receptor subfamily 4, group A, member 1 (*Nr4a1),* FBJ osteosarcoma oncogene *(Fos),* early growth response 1 *(Egr1),* neuronal PAS domain protein 4 *(Npas4),* and cysteine-rich protein 61 *(Cyr61)* [41]. The expression of IEGs is first induced in response to extracellular stimuli and is widely used as a marker of neural activity. Fos, Egr-1, and Npas4 play key roles in long-term synaptic plasticity [54]. In addition, Npas4 regulates the excitatory and inhibitory balance within circuits. Nr4a1 plays a role in regulating the density and distribution of spines and synapses [55], and in suppressing the age-related decline in brain function [56]. Cyr61 is needed for dendritic arborization in the hippocampus [57]. The increased expression of these IEGs in the hippocampus is thought to increase synaptic plasticity, contributing to the maintenance and improvement of learning and memory abilities. The transcription of many IEGs in neurons is initiated by the calcium ion influx associated with synaptic activity and action potential [58], and EGCG modulates calcium signals in hippocampal neurons [59,60]. Therefore, EGCG may increase the expression of IEGs via an increase in the levels of intracellular calcium ions in hippocampal nerve cells.

### 2.7. Actions of EGCG and Its Degradation Products on Brain

Since the transfer of catechins to the brain parenchyma is restricted by the BBB, the amount of catechins in the brain is considered to be lower than that in the periphery. The concentration-dependent effect of green tea catechins on cognitive function was observed in mice, but it was suggested that a daily intake of 1 mg/kg or more of green tea catechins suppresses age-related cognitive decline [41]. Epidemiological studies have also reported that a daily intake of one or several cups of green tea reduces the risk of dementia [3,4,42,43,44]. The quantity of catechins contained in green tea is generally around 70 mg/100 mL when eluted with hot water from tea leaves, and 30–50 mg/100 mL in commercially available bottles of green tea. Based on the BBB permeability of catechins, as discussed in Section 2.4, the concentration of EGCG and its degradation products will presumably promote the differentiation of nerve cells when several cups of green tea, or approximately one bottle of catechins, are consumed. This may reduce the age-related decline in cognitive function.

We hypothesize that by drinking a few cups of green tea every day, both EGCG and its degradation products may be taken up in the brain and activate nerve cells, as shown below and in Figure 4.
Two to three hours after ingesting green tea, a very small amount of EGCG is taken up from the small intestine and enters the brain parenchyma through the bloodstream.The incorporated EGCG activates the nerve cells by enhancing IEG expression and promoting cell differentiation. After this, the EGCG is excreted.On the other hand, in the large intestine, most of the EGCG is decomposed from EGC into valerolactones, such as EGC-M5, by the intestinal microbiota over a period of 8 h.The resulting EGC-M5 (and other decomposed products) is taken up from the large intestine.EGC-M5 and its conjugates are delivered to the brain via systematic circulation.Nerve cells are reactivated again.

In this way, by drinking a few cups of green tea every day, brain aging might be slowed. The bioavailability of catechins was thought to be low, but in fission metabolites, it has been reported to be approximately 40 to 62%, albeit with large variability among individuals [52,61]. It has been pointed out that phenylvalerolactone and phenylvaleroic acid are important as metabolic fission products common to flavan-3-ols [52]. In addition to the effects of EGCG degradation products on nerve cells, beneficial effects on blood pressure, immunity, and glucose metabolism have been reported [18,19,20]. The physiological effects of catechin degradation products will become even more apparent in the future.

## 3. Theanine and Arginine

Theanine is one of the amino acids that has been discovered in Gyokuro, a high-grade green tea. In 1950, Yajiro Sakato revealed its chemical structure to be γ-glutamylethylamide [62]. Theanine is an amino acid unique to tea. Almost no theanine is detected in other kinds of Camellia (Genus *Camellia*) plants [63]. Amino acids make up 1–8% of dried tea leaves, of which theanine makes up approximately half [64]. Since theanine is synthesized in the roots of tea and stored in the leaves via the trunk, it is thought that theanine is produced to store the absorbed ammonia nitrogen in a safe form [65]. The young shoots at the tip have a high content of theanine. Because theanine is metabolized into polyphenols such as catechins in sunlight [66], blocking the sunlight for two to three weeks before harvesting suppresses the decomposition of theanine and maintains high levels. Arginine is the second most abundant free amino acid in tea cultivated in Japan after theanine, but its content is low in Assam and its hybrids [67].

As theanine is similar to glutamic acid, which is one of the transmitters in the brain, it was predicted to have some physiological action in the brain, and many studies have been conducted to address this. Theanine absorbed from the intestine enters the brain through the BBB [68]. Furthermore, it has been reported that theanine’s effects on neurotransmitters in the brain, such as dopamine [66], suppress excitation by caffeine [69], improve memory [70], have a high affinity for the glutamine transporter [70], and enact a neurogenic effect [71]. As regards the actions of theanine in humans, a relaxing effect [26], stress reduction [24,25], and a reduction in depression and schizophrenia [72] have been reported. On the other hand, little attention has been paid to the functionality of arginine in green tea, but it has been revealed to have an excellent stress-reducing effect, similar to that of theanine [27,28]. Here, we focus on the suppression of brain aging via the stress-reducing effects of theanine and arginine and introduce a partially elucidated mechanism.

### 3.1. Adrenal Hypertrophy by Psychosocial Stress and Its Suppression by Theanine

Moderate stress is thought to be necessary and beneficial, but when excessive stress is experienced for a long period of time, it causes the onset and exacerbation of various diseases, such as depression, mood disorders, cardiovascular diseases, and age-related diseases [73,74]. A confrontational housing situation, using territorial male mice, increases the psychosocial stress load in each mouse, which is similar to the stress encountered by humans (Figure 5) [75]. When the mice experience stress, stress responses, such as changes in hormone secretion and adrenal hypertrophy, are observed in the activation of the hypothalamus–pituitary–adrenal (HPA) axis via excitatory signals. Adrenal hypertrophy and the altered diurnal rhythm of glucocorticoids were observed in mice placed in confrontational housing. However, the ingestion of theanine (6 mg/kg) brought them to a normal state [75]. The normal circadian rhythm of glucocorticoids has been reported to be important for synaptogenesis in the brain [76], and stress-induced hormonal disturbances are thought to cause cognitive decline. The normalization of the HPA axis by theanine may also affect brain function.

Adrenal hypertrophy, due to confrontational housing, has been observed in all strains of male mice examined so far, but differences in resilience to stress have been found. For example, adrenal hypertrophy began to decrease after 10 days of stress loading in ddY mice [75] (a commonly used outbred strain), but hypertrophy was observed even after 7 months in senescence-accelerated SAMP10 mice [77]. SAMP10 mice have a shorter lifespan than normal mice, and brain atrophy and decreased brain function are observed with aging [77]. Theanine showed an excellent suppressive effect on adrenal hypertrophy in mice of both strains. In females, adrenal hypertrophy was not clearly observed under confrontational housing conditions, because females are less territorial than males.

### 3.2. Longevity Shortened by Stress and Its Suppression by Theanine and Arginine

The average survival time of SAMP10 mice was 17.6 ± 1.2 months under normal group housing conditions, which decreased significantly under confrontational housing conditions to 13.6 ± 1.5 months (Figure 6) [77]. Survival was reduced by three quarters due to stress loading. However, mice drinking water containing theanine (6 mg/kg) under the same stress-loading conditions showed a similar survival time to group-housed mice (17.9 ± 1.4 months). The longevity of group-housed mice was not prolonged by theanine intake, indicating that theanine suppresses the shortening of survival time by reducing stress. On the other hand, no shortening of lifespan due to stress load was observed in ddY mice. Individual differences in susceptibility to stress are well known, but it is not yet fully understood what causes these differences. SAMP10 and ddY mice may be suitable laboratory animals to assess differences in stress sensitivity.

The median survival time (MST) was significantly longer in SAMP10 mice ingesting arginine (3 mg/kg) under stress-loading conditions (16.6 months) than in mice receiving no arginine (10.5 months) [28]. The dose of arginine was based on data regarding the suppression of adrenal hypertrophy, which was observed to occur at approximately half the dose of theanine. It was also determined by the fact that arginine represents approximately half the theanine in green tea.

### 3.3. Promotion of Cognitive Decline Due to Stress Load, and Accumulation of Oxidative Damage

SAMP10 mice undergo a significant reduction in learning ability after 11 months of age, but no reduction has yet been observed at 8 months of age. However, under stress-loading conditions, it has been found that learning ability declines as early as 8 months of age, indicating that stress promotes a decline in brain function [77]. On the other hand, no reduction in brain function has been observed in stress-loaded mice ingesting theanine or arginine [28,77]. Since the brain consumes a large amount of oxygen, it is susceptible to oxidative damage due to the production of many reactive oxygen species (ROS) during the metabolic process [78]. When the level of 8-oxodeoxyguanosine in cerebral cortical DNA was measured at 9 months of age as a marker of oxidative damage, this damage was seen to be significantly increased in mice under stress-loaded conditions as compared with mice of the same age under group housing conditions [77]. SAMP10 mice produce more ROS in the brain from a young age than normal mice [79], and the activity of glutathione peroxidase tends to decrease in aged mice [80]. Therefore, oxidative damage tends to accumulate with age. However, as theanine has no direct or strong antioxidant effect similar to catechins, the suppression of oxidative damage via the ingestion of theanine is considered to be indirectly caused by the balance of ROS production/elimination in the brain.

### 3.4. Brain Atrophy Due to Stress Load and Suppression by Theanine

It has been reported that severely stressed people [30,31,32] and abused children [33,34,35] experience brain atrophy. SAMP10 mice show cerebral atrophy with aging, and it was revealed that stress load further promotes cerebral atrophy [77]. In order to clarify when and where stress-induced brain atrophy occurs, the brains of mice under confrontational housing conditions were closely analyzed using ex vivo magnetic resonance scanning (MR) [81]. The mice were kept under confrontational housing conditions. The theanine group freely drank water containing theanine (6 mg/kg/day). The control mice drank water. Significant atrophy occurred in the cerebral cortex one month after stress loading, and then atrophy progressed in the control mice (Figure 7a) [81]. Similar atrophy levels were observed in the theanine group mice one month later under stress loading, but these were recovered after 2 months. The hippocampus tended to atrophy one month after the start of stress loading, but at 6 months, the hippocampus of mice in the theanine group was significantly larger than that of control mice [81], while the cerebral cortex of ddY mice atrophied in the control group one month after the start of stress loading but recovered thereafter [81] (Figure 7b). No atrophy was observed in the theanine group of ddY mice. Similar effects were observed in the hippocampus. There was a difference in stress-induced brain atrophy between SAMP10 and ddY mice. These results confirm that stress-induced atrophy of the brain occurs at an early stage, and theanine is involved in the suppression and recovery of stress-induced brain atrophy.

### 3.5. Changes in Hippocampal Gene Expression Due to Stress Loading and Suppression by Theanine or Arginine

Since changes in the initial stage of stress loading were clearly important, changes in hippocampal gene expression on the third day of stress loading were comprehensively investigated. The expression of lipocalin 2 (*Lcn2*) was markedly increased in SAMP10 mice by stress loading but was clearly suppressed following theanine intake [81] (Figure 8). The increase in ddY was slight. Excessive Lcn2 expression has been reported to lead to increased neuronal cell death, as Lcn2 is released from activated astrocytes and increases nerve cell damage and inflammation [82]. Excessive Lcn2 expression in SAMP10 mice is thought to be a primary cause of brain atrophy. In addition, the control of Lcn2 by theanine may be critical in suppressing the chronic inflammation associated with aging and neurodegenerative diseases.

On the other hand, the expression of Npas4 was significantly reduced in the hippocampus of SAMP10 mice by stress loading, but this reduction was suppressed in the mice ingesting theanine [81] (Figure 8). In ddY mice, the expression of Npas4 was increased, while this was slightly suppressed in the theanine group. Npas4, one of the IEGs, plays an important role in neural activity-dependent synaptogenesis as a transcription factor and is closely involved in anxiety, depressive behavior, and learning behavior [83]. It has been reported that Npas4 expression in the hippocampus decreases when rats are subjected to chronic stress, but this level is more effectively restored in stress-resistant (resilient) compared to stress-vulnerable rats [84]. These early changes in gene expression are considered to be one reason for the stress vulnerability of SAMP10 mice. Although SAMP10 is vulnerable to stress, it has been suggested that theanine may enhance resilience by regulating Npas4 and Lcn2 expression.

Glutamate, the major excitatory neurotransmitter, regulates the HPA axis via glutamate receptors in the paraventricular nucleus of the hypothalamus [85]. Theanine, taken into the brain, is reported to act strongly on the glutamine transporter and suppress hyperexcitability by suppressing the supply of glutamate [70]. Since Npas4 also regulates nerve excitement/inhibition, it is necessary to investigate how the balance between glutamate and γ-aminobutyric acid (GABA), an inhibitory neurotransmitter, is altered by theanine intake.

Next, to investigate the mechanism of the anti-stress effect of arginine, early changes in gene expression levels were measured in SAMP10 mice on day three of stress loading [28]. The genes that maintain mitochondrial functions and neuronal survival, including hemoglobin alpha, adult chain 2 (*Hba-a2*) and hemoglobin beta, adult minor chain (*Hbb-b2*), were significantly increased in mice that ingested arginine. In contrast, several genes associated with the oxidative stress response and neuronal excitotoxic cell death, including nuclear receptor subfamily 4, group A, member 1 (*Nr4a1*), activity-regulated cytoskeleton-associated protein (*Arc*), and cysteine-rich protein 61 (*Cyr61*), were remarkably increased in response to stress. However, their expression was significantly suppressed in mice that ingested arginine. In fact, lipid peroxidation in the brains of stressed mice was significantly lowered by arginine intake [28]. These results indicate that arginine reduces oxidative damage and enhances mitochondrial functions in the brain.

Both theanine and arginine have a stress-reducing effect, but there are differences in the major genes that they each target. Since many changes occur in organisms in response to stress loading, it is possible that there are various targets in stress reduction.

### 3.6. Changes in Hippocampal Metabolites and Behavior One Month after Stress Loading

Furthermore, the effects of theanine intake on behavior were compared at one month after stress loading, at which point cerebral atrophy was markedly observed. As a result, in SAMP10 and ddY mice, the immobility time in the tail suspension test was significantly shortened in the theanine intake group, suggesting that theanine intake improves depressive behavior in stress-loaded mice [86]. After this, changes in the metabolites in the hippocampus were investigated. The hippocampus is a stress-sensitive tissue in the brain. For example, chronic stress has been shown to significantly reduce the hippocampal volume in mice and impair hippocampal neurogenesis [87]. Neurogenesis in the hippocampus occurs throughout life, which regulates the inhibitory circuitry [88] and could be associated with hippocampus-dependent learning and memory [89,90,91].

Among the statistically altered metabolites, kynurenine (Kyn) and histamine were significantly higher in SAMP10 than in ddY. Kyn has been reported to increase in chronically stressed rats [92]. In addition, the Kyn pathway plays a key role in depressive behavior in mice [93,94]. Since Kyn is produced from tryptophan via indoleamine-2,3-dioxygenase (IDO), the high expression level of IDO in SAMP10 mice under stress loading may contribute to the high level of Kyn in SAMP10 mice [86]. Histamine, which is enhanced by a variety of stressors, has strong effects on excitability in the hippocampus [95]. The synthesis of histamine is controlled by the inhibitory H3 auto-receptors located on histamine neurons [96]. Theanine intake significantly suppresses histamine levels in stress-loaded SAMP10 mice [86], suggesting that the histaminergic system is an important target for theanine. In addition, theanine has been suggested to improve sleep quality [72,97]. It is interesting that histamine likely plays a pivotal role in the regulation of sleep–wakefulness via the H1 and/or H3 receptors [96].

On the other hand, carnosine has been shown to be present at high concentrations in the brain and has been reported to have antidepressant effects [98,99]. Carnosine levels were significantly lower in SAMP10 than ddY mice, but these levels were significantly increased in SAMP10 mice that ingested theanine under confrontational housing conditions [86]. In addition, the levels of ornithine, which has anti-stress effects, were significantly lower in SAMP10 mice than in ddY mice [86].

In summary, depressive behavior was observed under long-term stress conditions. Here, depression-related Kyn and excitement-related histamine levels increased in the hippocampus, while antidepressant-related carnosine and anti-stress-related ornithine levels decreased. It has been suggested that theanine intake regulates their levels and improves depressive behavior. These metabolic differences may help to determine stress vulnerability.

### 3.7. Brain Aging Acceleration Due to Psychosocial Stress

The relationship between stress and brain aging described in this section can be summarized as follows (Figure 9). In stress-vulnerable mice (SAMP10), brain atrophy was observed at least one month after stress loading, which then further progressed with aging, resulting in aging promotion (decreased brain function and shortened lifespan). On the other hand, in stress-resilient mice (ddY), the degree of atrophy in the brain due to the stress load was not significant; furthermore, this atrophy was recovered, and no effect on brain function or longevity was observed.

In SAMP10 mice given theanine, stress-induced atrophy in the brain subsequently recovered, though not completely. No atrophy was observed in ddY mice receiving theanine. In addition, mice ingesting theanine and arginine exhibited suppressed cognitive decline and lifespan shortening. Stress seems to cause neuroinflammation, oxidative stress, imbalances in excitation/suppression, and the suppression of neurogenesis in the brain. Although some of theanine and arginine’s target genes have been identified in the early stages of stress loading, further research is needed to elucidate the effects of theanine and arginine on complex biological stress responses. Nevertheless, for those who are vulnerable to stress, theanine and arginine are probably essential substances.

## 4. Function of Green Tea

Green tea (*Camellia sinensis* (L.) O. Kuntze) is an unfermented tea that remains green via the rapid inactivation of enzymes during heating, such as polyphenol oxidase, and the prevented oxidation of the components. There are two main types of green tea [100]. One is Sencha, which is made from ordinary tea leaves and is rich in catechins. The other includes Gyokuro and Tencha, which are deprived of sunlight for approximately 20 days before harvesting. Shaded tea leaves have a high content of theanine because their metabolization into catechins is suppressed. The theanine content is higher in higher-grade green tea. Matcha tea is made by grinding Tencha into a fine powder with a stone mill. Catechins, caffeine, and amino acids containing theanine provide the taste components of astringency, bitterness, and umami, respectively. In addition, the balance of these ingredients affects the functionality of the tea.

### 4.1. Stress-Reducing Effect of Green Tea with Lowered Caffeine

It has been determined that the stress-reducing effect of theanine is strongly suppressed by the presence of caffeine and EGCG [27]. Therefore, the amount of caffeine in picked tea leaves can be reduced to 1/3–1/4 by spraying them with hot water [27]. The tea produced in this way is called low-caffeine green tea.

The elution of caffeine and EGCG depends on the water temperature—their elution degree decreases in cold water—but amino acids and EGC are less affected by water temperature [101]. Therefore, low-caffeine green tea eluted with cold water has low levels of caffeine and EGCG and relatively high levels of theanine, arginine, and EGC. The mean value of each component of green tea used in clinical studies with participants in their 80s–90s is shown in Table 2.

Participants in their 20s (young), 40s–50s (middle-aged), and 80s–90s (elderly) drank low-caffeine green tea eluted with room-temperature water. Salivary alpha-amylase is a non-invasive biomarker of the sympathetic nervous system [102]. The salivary amylase activities (assessed in the subjects waking up in the morning) in the low-caffeine green tea intake group were shown to be lower than those in the control group (standard green tea or barley tea) in all age groups [103,104,105]. This indicates that the ingestion of low-caffeine green tea is useful for suppressing stress. A self-diagnostic assessment of accumulated fatigue and work severity, which was created by the Japanese Ministry of Health, Labor, and Welfare, was carried out every Monday morning and every Friday evening throughout the test period. The degree of fatigue accumulation on Monday morning was significantly lower in the middle-aged low-caffeine green tea intake group [104]. An electroencephalogram (EEG) was taken during sleep using a single-channel EEG (Sleep Scope, SleepWell Co., Osaka, Japan). Although there were individual differences in sleep characteristics, an early-morning awakening was suppressed in the elderly low-caffeine green tea intake group [105]. A similar effect was observed in middle-aged people [104]. Early-morning awakening, with an inability to return to sleep, is a common sleep problem in older people [106], and so the intake of low-caffeine green tea may improve their quality of sleep. In addition, in elderly participants whose salivary amylase activity was lowered by the ingestion of low-caffeine green tea, several sleep parameters, such as sleep efficiency, sleep onset latency, and non-REM sleep, were improved [105]. These data show that reducing the caffeine and EGCG levels in green tea not only enhances the stress-reducing effects of theanine but also improves sleep. Better sleep is broadly beneficial because chronic insomnia can increase the risk of cognitive impairment, anxiety, depression, and cardiovascular disease [107].

### 4.2. Stress Reduction by Matcha

Matcha is a green tea that contains more theanine and caffeine, and fewer catechins, than Sencha. However, these components vary considerably depending on the cultivation conditions and harvest time. Therefore, some samples of Matcha were evaluated for their stress-reducing effects via an animal experiment. Here, Matcha with a CE/TA ratio of 2 or less—where the CE/TA ratio indicates the molar ratio of the sum of caffeine (C) and EGCG (E) to the sum of theanine (T) and arginine (A)—was able to suppress stress, indicated by adrenal hypertrophy, in mice [29].

The stress-reducing effect of Matcha suspended in water was examined in participants in their 20s. When evaluated by salivary amylase activity, Matcha with a CE/TA ratio of 1.79 was shown to reduce stress, but that with a CE/TA ratio of 10.79 did not show any effect [29]. A similar difference in effect was observed when these Matcha varieties were baked into cookies and ingested [108].

Of the 76 brands of Matcha sold in Japan, 32 had a CE/TA ratio of 2 or less [29] (Figure 10). Of the 67 brands of Matcha sold outside of Japan, only one brand had a CE/TA ratio of 2 or less. These data indicate that not all Matcha varieties can be expected to have a stress-relieving effect.

### 4.3. Anti-Depressant Effect of Shaded White Leaf Tea (SWLT)

When a green tea plant is completely shaded for approximately two weeks, its levels of amino acids can be increased six- to seven-fold as compared with ordinary Sencha. The green tea produced in this way is called shaded white leaf tea (SWLT), and it is gaining interest as a green tea with a strong umami taste [109]. In SWLT, the theanine level is increased around 5-fold, and the arginine around 13-fold, compared to normal Sencha (Table 3); its CE/TA ratio is 1.12, thus suggesting a stress-reducing effect. However, no significant stress-reducing effect of this variety was observed in a clinical study [110].

In SWLT, complete shading for two weeks causes the decomposition of soluble proteins, and its free amino acid composition is significantly different from that of general Sencha [110]. Although theanine levels are increased in this variety, the proportion of theanine in its total amino acid level is lower compared to Sencha, while the proportions of arginine, glutamine, asparagine, aspartic acid, etc., were higher. An increase in asparagine and aspartic acid was found to suppress the anti-stress effect of theanine [110]. Furthermore, no stress-reducing effect was observed in the eluate with a CE/TA ratio of 0.9 or higher. It seems that tea eluate and Matcha are somewhat different.

On the other hand, depressive behavior was significantly suppressed in mice ingesting SWLT for one month [110]. Depression is the most common mental illness, and stress is a major risk factor for it. Stress loads initially cause excitement, but chronic stress leads to depression. Therefore, in the context of long-term stress, a green tea that not only reduces stress but also alleviates depression is necessary. The intake of green tea has been reported to prevent depression, suggesting the involvement of caffeine and catechins [111]. For example, caffeine and EGCG have been reported to increase the expression of glutamate, a major excitatory neurotransmitter in the brain, while inhibiting GABA, an inhibitory neurotransmitter [112,113]. Balancing excitement and depression in the brain is considered to be very important for maintaining physical and mental health. In the near future, it will be possible to produce green tea with a CE/TA ratio suitable for improving depression in humans.

## 5. Conclusions

Epidemiological and animal studies have suggested that daily intake of green tea catechins suppresses age-related cognitive decline. EGCG, the main catechin in green tea, has been suggested to activate nerve cells, and its metabolic decomposition products act similarly, with a time lag. Based on the BBB permeability of catechins and their degradation products, several cups of green tea, or approximately one bottle of catechins, may reduce the age-related decline in cognitive function.

In addition, theanine and arginine were shown here to have an excellent stress-reducing effect and to suppress the shortening of lifespan and the deterioration in cognitive function due to stress. Since brain atrophy followed by aging promotion, due to psychosocial stress, were observed in stress-vulnerable mice, but not in stress-resistant ones, theanine and arginine are probably essential substances for individuals who are vulnerable to stress.

However, the effect of theanine and arginine was shown to be greatly determined by the simultaneous presence of caffeine and EGCG. Differences in the molar ratio of CE/TA (caffeine + EGCG/theanine + arginine) in green tea were shown to affect stress reduction and sleep in experimental and clinical studies.

EGCG and theanine are unique ingredients in green tea that affect brain function. The daily intake of a quantity of green tea appropriate for each physical and mental health condition is suggested to suppress brain aging by activating nerve cells and reducing stress.

## Figures and Tables

**Figure 1 molecules-26-04897-f001:**
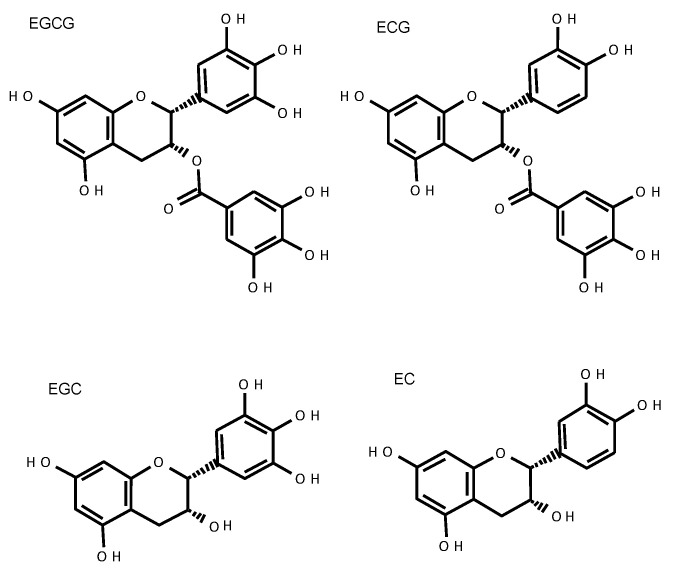
The structure of green tea catechins.

**Figure 2 molecules-26-04897-f002:**
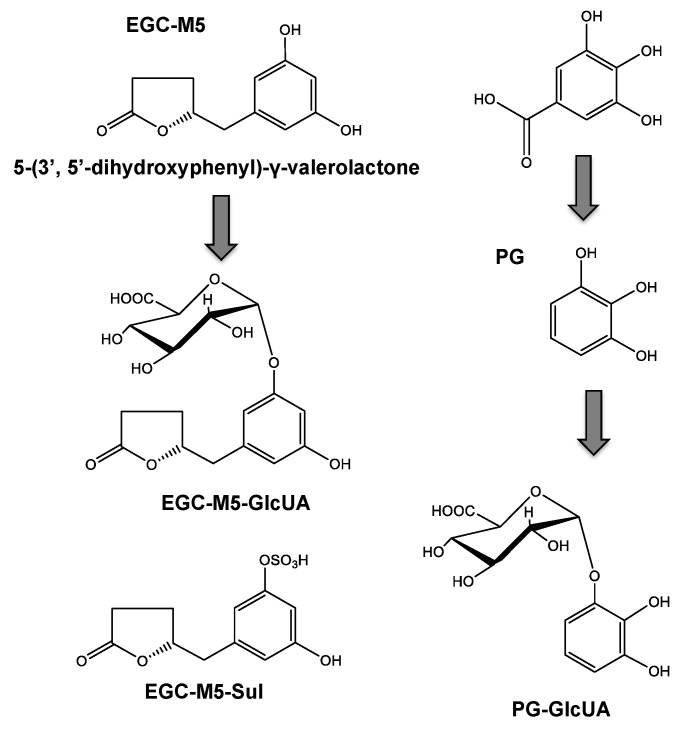
The structure of EGCG metabolites.

**Figure 3 molecules-26-04897-f003:**
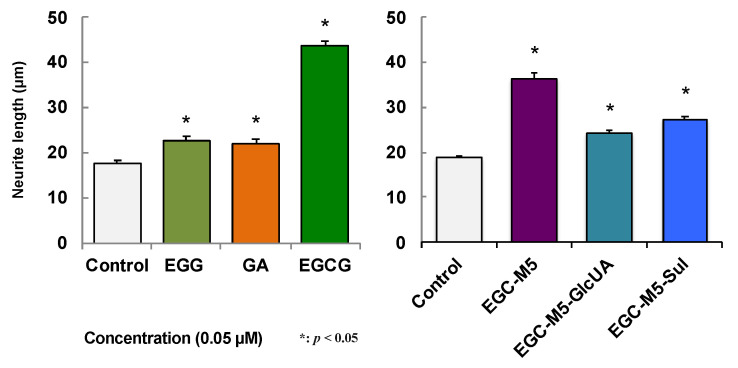
The neurite outgrowth of human SH-SY5Y cells treated with EGCG and its metabolites. A cell suspension (2.5 × 10^4^ cells/well) was plated in a 24-well plate. Samples dissolved in 0.01% DMSO were added to the culture medium to produce a final concentration of 0.05 μM, and these were cultured for 72 h at 37 ℃. Each value represents the mean ± SEM (neurite length, n = 103–107). Asterisks represent significant differences relative to the control (* *p* < 0.05, Bonferroni’s *t*-test) [17,48].

**Figure 4 molecules-26-04897-f004:**
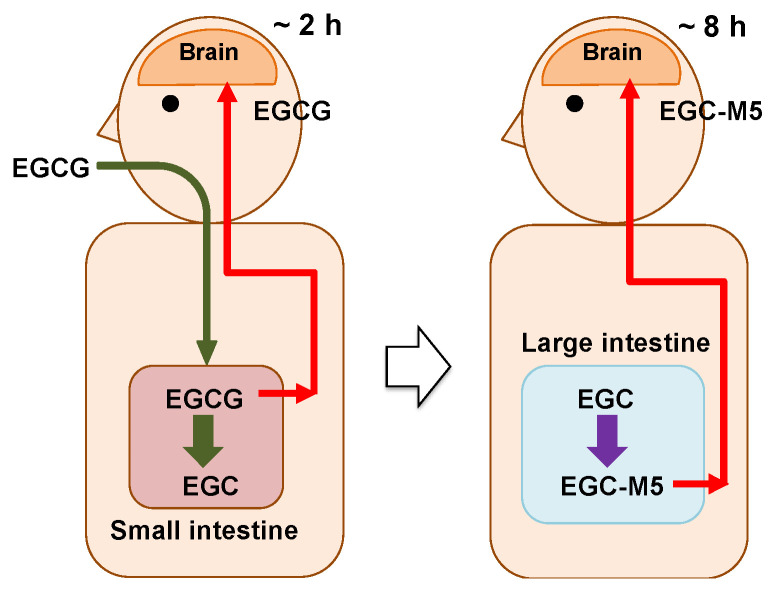
The hypothesis that EGCG and its degradation products act on the brain with a time lag.

**Figure 5 molecules-26-04897-f005:**
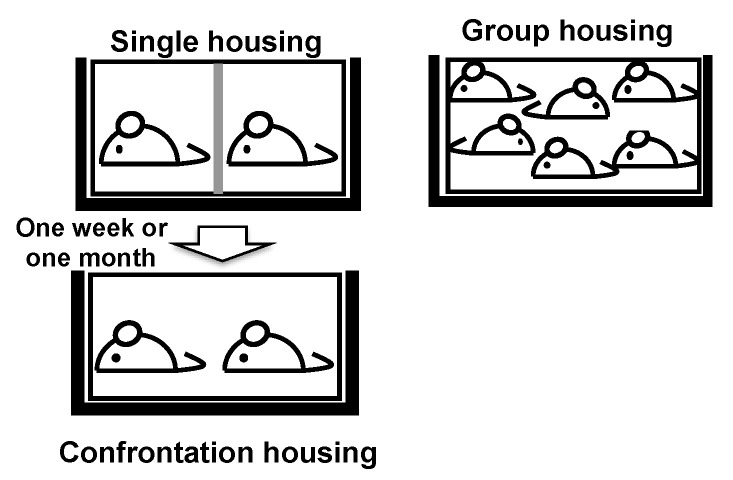
Confrontational housing as a psychosocial stress loading scenario. To create confrontational housing, a standard polycarbonate cage was divided into two identical subunits with a stainless-steel partition (single housing). Two mice were housed in the partitioned cage for one week (for a short-term experiment) or one month (for a long-term experiment). Then, the partition was removed, and, subsequently, the two mice co-inhabited the same cage (confrontational housing). Group-housed mice were housed in groups of six as a control housing scenario. These mice ingested water with or without theanine (6 mg/kg).

**Figure 6 molecules-26-04897-f006:**
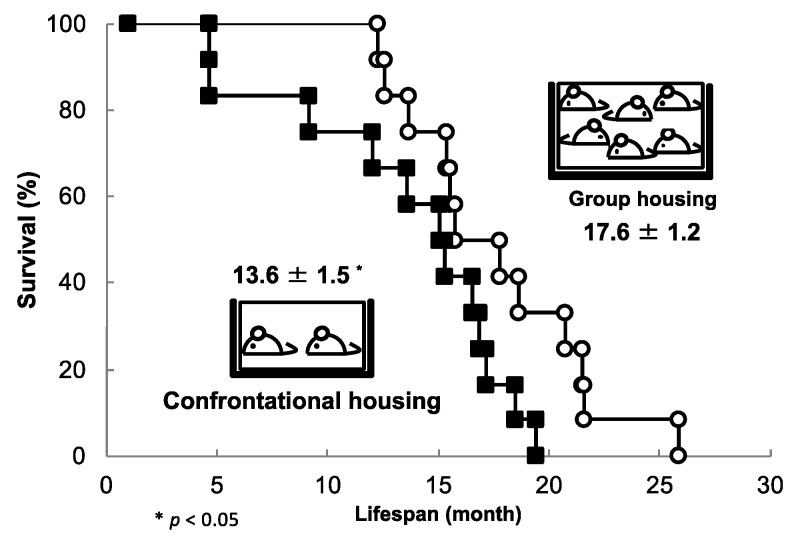
The effect of confrontational housing on the lifespan of SAMP10 mice. Mice were housed alone for one month before confrontational housing conditions were induced. Mice that died early were observed, and the maximum survival time was approximately 20 months (closed square). In contrast, group-housed control mice began to die at around 12 months of age, with a maximum survival time of 26 months (open circles) (n = 12) [77].

**Figure 7 molecules-26-04897-f007:**
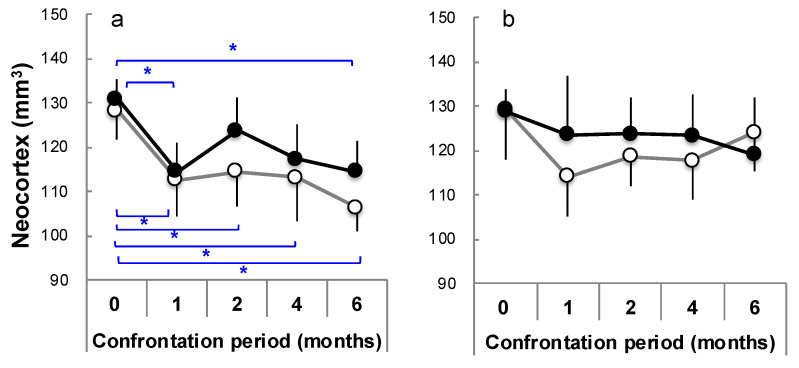
The time course of the neocortex volume in SAMP10 (**a**) and ddY mice (**b**) between the control (open circle) and theanine (closed circle) groups under confrontational housing conditions. The mice were dissected after 0, 1, 2, 4, and 6 months of confrontational housing and subjected to measurement for ex vivo MR (n = 3–12; *, *p* < 0.05) [81].

**Figure 8 molecules-26-04897-f008:**
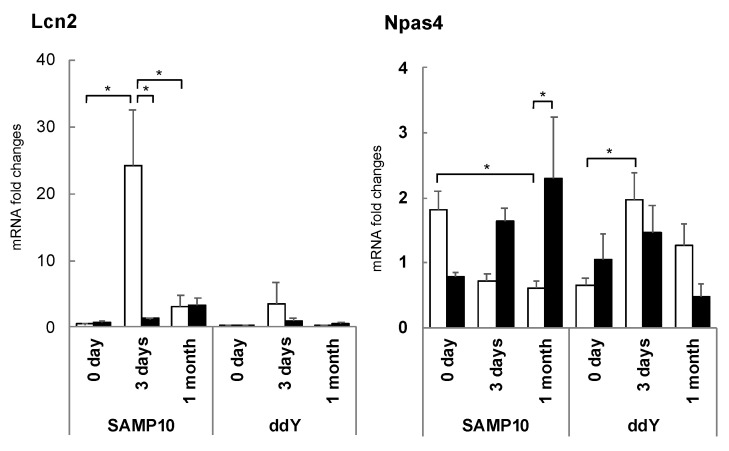
Expression levels of Lcn2 and Npas4 mRNA in the hippocampus of SAMP10 and ddY mice. Mice consumed theanine (closed bar) or normal tap water (control, open bar) ad libitum. After single housing for one month, hippocampal samples were obtained from mice housed confrontationally after 0 days, 3 days, and 1 month. Values are expressed as the means ± SEM (n = 3–6, * *p* < 0.05) [81].

**Figure 9 molecules-26-04897-f009:**
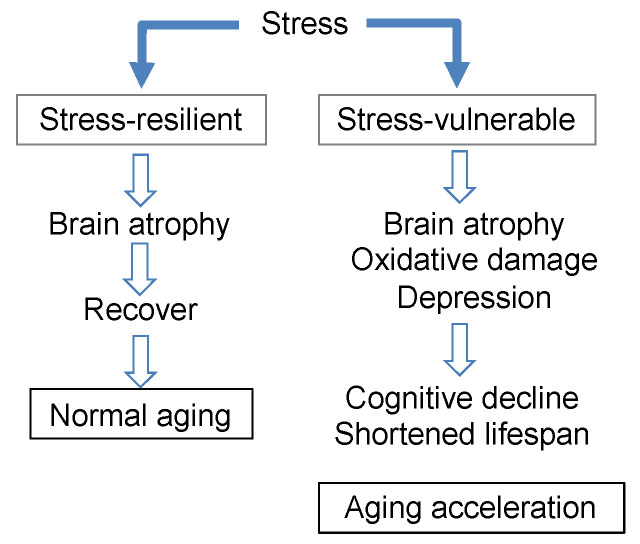
The relationship between stress and brain aging.

**Figure 10 molecules-26-04897-f010:**
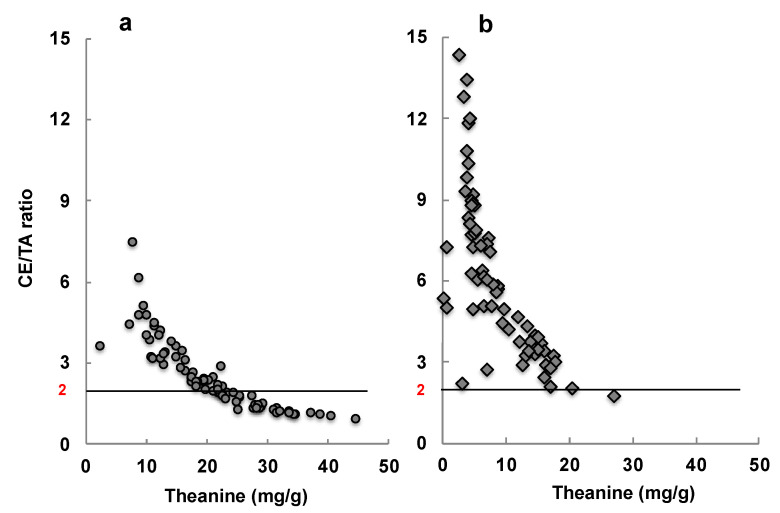
The amount of theanine, and the molar ratio of caffeine and EGCG to theanine and arginine, in 76 samples of Matcha marketed in Japan (**a**) and 67 samples of Matcha marketed outside of Japan (**b**).

**Table 1 molecules-26-04897-t001:** BBB permeability of green tea catechins and their metabolites.

Sample	Coexistence Sample	Permeability Coefficient (10^−6^ cm/s)	BBB Permeability (%) (0.5 h)
EGCG	−	13.45 ± 0.57	4.00 ± 0.17
EGCG	EGC	10.53 ± 0.63	3.13 ± 0.19
EGC	−	16.70 ± 1.86	4.96 ± 0.55
EGC	EGCG	12.39 ± 1.56	3.68 ± 0.47
EGC	GA	6.01 ± 1.57	1.79 ± 0.47
GA	−	31.73 ± 3.39	9.42 ± 1.01
GA	EGC	26.98 ± 2.77	8.01 ± 0.82
EGC-M5	−	17.99 ± 0.79	5.34 ± 0.23
EGC-M5-Sul	−	14.61 ± 1.35	4.34 ± 0.40
EGC-M5-GlcUA	−	12.53 ± 0.02	3.72 ± 0.01
Pyrogallol	−	13.79 ± 1.62	4.10 ± 0.48
Pyro-GlcUA	−	9.28 ± 1.41	2.76 ± 0.42
Caffeine	−	45.22 ± 4.41	13.43 ± 1.31

Data are expressed as the mean ± SEM (n = 3). (Data of Ref. 17 and 48 are modified).

**Table 2 molecules-26-04897-t002:** The content of caffeine, catechins, and amino acids in a solution of green tea.

**Green Tea**	**Caffeine (mg/L)**	**Catechins (mg/L)**								
		**EGCG**	**EGC**	**ECG**	**EC**	**CG**	**(+) C**	**Total**		
Standard	120	117	73.2	26.5	43.9	0.41	5.47	266		
Low -caffeine	37.2	64.7	145	10.7	53.8	0.26	4.13	278		
**Green tea**	**Free amino acids (mg/L)**									
	**Theanine**	**Glu**	**Arg**	**Asp**	**Gln**	**Ser**	**Ala**	**Asn**	**GABA**	**Total**
Standard	36.0	9.12	7.28	7.44	5.34	2.99	0.90	0.76	0.47	70.4
Low -caffeine	85.2	18.2	21.6	12.9	15.8	5.73	2.16	1.21	1.51	164

Low-caffeine green tea (20 g) was steeped in 2000 mL room-temperature water. Standard green tea (10 g) was steeped in 2000 mL boiling water. EGCG, (−) epigallocatechin gallate; EGC, (−) epigallocatechin; ECG, (−) epicatechin gallate; EC, (−) epicatechin; CG, (−) catechin gallate; (+) C, (+) catechin; Glu, glutamic acid; Arg, arginine; Asp, aspartic acid; Gln, glutamine; Ser, serine; Ala, alanine; Asn, asparagine; GABA, γ-amino butyric acid.

**Table 3 molecules-26-04897-t003:** The caffeine, catechin, and amino acid contents in the eluate of SWLT and Sencha.

**Tea**	**Caffeine (mg/L)**	**Catechins (mg/L)**							
		**EGCG**	**EGC**	**ECG**	**EC**	**GC**	**CG**	**(+) C**	
SWLT	209.8	150.4	135.2	24.6	41.0	5.0	2.8	3.4	
Sencha	112.0	134.2	229.0	21.0	46.6	13.6	4.6	2.0	
**Tea**	**Free amino acids (mg/L)**								
	**Theanine**	**Arg**	**Gln**	**Asn**	**Asp**	**Glu**	**Ser**	**GABA**	**Total**
SWLT	140.2	69.9	51.7	33.8	33.5	19.3	12.6	0	361.0
Sencha	28.8	5.4	3.9	0.7	5.5	6.9	2.2	0	53.5

Shaded white leaf tea (SWLT) and Sencha green tea (3 g) were steeped in 500 mL room-temperature water for 3 h.

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
