# Peer review of "Green Tea Suppresses Brain Aging"

_molecules, 2021, doi:10.3390/molecules26164897_

Round 1
Reviewer 1 Report
As far as my skills are concerned, I have no critical observations to make.
Author Response
As far as my skills are concerned, I have no critical observations to make.
Thank you very much for reviewing our manuscript.
Reviewer 2 Report
The presented review is a comprehensive study that summarizes the knowledge regarding the effect of green tea in the activation of nerve cells and suppressing brain aging.
I find this review interesting, scientifically justified, well written and detailed. The literature is up to date and thoroughly investigated. The review represents an original, complete and well-planned research. My overall impression is that the authors did a great job in the research of bioactive compounds.
However, some minor issues can be addressed in order to further improve the clarity of the manuscript:
The Conclusion section should be rewritten to highlight important conclusions
There are some minor typographical and grammatical errors throughout the manuscript that should be checked and corrected.
Author Response
The presented review is a comprehensive study that summarizes the knowledge regarding the effect of green tea in the activation of nerve cells and suppressing brain aging.
I find this review interesting, scientifically justified, well written and detailed. The literature is up to date and thoroughly investigated. The review represents an original, complete and well-planned research. My overall impression is that the authors did a great job in the research of bioactive compounds.
However, some minor issues can be addressed in order to further improve the clarity of the manuscript:
The Conclusion section should be rewritten to highlight important conclusions
Thank you so much for reviewing our manuscript. We are deeply grateful for your valuable suggestions. We revise the conclusion section to highlight important conclusions.
There are some minor typographical and grammatical errors throughout the manuscript that should be checked and corrected.
We asked MDPI English Service again to check the revised manuscript.
Reviewer 3 Report
In the present manuscript, entitled ”Green Tea Suppresses Brain Aging” the authors focused on the effects of green tea components - catechins, EGCG metabolites, theanine and arginine - on brain function. They explained the anti-stress and antidepressant effects of several types of green tea. The authors described that green tea suppresses brain aging, through the activation of nerve cells by both EGCG and its degradation products, and the reductions in stress by theanine and arginine.
The prevention of brain aging is a universal problem.
Considering that dementia is affected by various factors, such as environment, lifestyle behavior, and brain aging, the improvement of the mentioned factors may delay the onset of dementia. Therefore, this review paper topic is of interest.
The manuscript is well-structured, well-written and easy to read.
The information is precisely and clearly presented, thus the non-expert readers can easily understand the message. Half of the references date from the past 5 -6 years.
In my opinion, the manuscript is certainly useful to the scientific community and deserves publication.
Author Response
In the present manuscript, entitled ”Green Tea Suppresses Brain Aging” the authors focused on the effects of green tea components - catechins, EGCG metabolites, theanine and arginine - on brain function. They explained the anti-stress and antidepressant effects of several types of green tea. The authors described that green tea suppresses brain aging, through the activation of nerve cells by both EGCG and its degradation products, and the reductions in stress by theanine and arginine.
The prevention of brain aging is a universal problem.
Considering that dementia is affected by various factors, such as environment, lifestyle behavior, and brain aging, the improvement of the mentioned factors may delay the onset of dementia. Therefore, this review paper topic is of interest.
The manuscript is well-structured, well-written and easy to read.
The information is precisely and clearly presented, thus the non-expert readers can easily understand the message. Half of the references date from the past 5 -6 years.
In my opinion, the manuscript is certainly useful to the scientific community and deserves publication.
Thank you so much for reviewing our manuscript. We really appreciate your support.